# SMARCA4 Mutations in Gastroesophageal Adenocarcinoma: An Observational Study via a Next-Generation Sequencing Panel

**DOI:** 10.3390/cancers16071300

**Published:** 2024-03-27

**Authors:** Kohei Yamashita, Matheus Sewastjanow-Silva, Katsuhiro Yoshimura, Jane E. Rogers, Ernesto Rosa Vicentini, Melissa Pool Pizzi, Yibo Fan, Gengyi Zou, Jenny J. Li, Mariela Blum Murphy, Qiong Gan, Rebecca E. Waters, Linghua Wang, Jaffer A. Ajani

**Affiliations:** 1Departments of Gastrointestinal Medical Oncology, University of Texas MD Anderson Cancer Center, Houston, TX 77030, USA; kyamashita@mdanderson.org (K.Y.); msewastjanow@mdanderson.org (M.S.-S.); ky@hama-med.ac.jp (K.Y.); ernesto.vicentini@gmail.com (E.R.V.); mppizzi@mdanderson.org (M.P.P.); yfan3@mdanderson.org (Y.F.); gzou@mdanderson.org (G.Z.); jjli2@mdanderson.org (J.J.L.); mblum1@mdanderson.org (M.B.M.); 2Department of Pharmacy Clinical Programs, University of Texas MD Anderson Cancer Center, Houston, TX 77030, USA; jerogers@mdanderson.org; 3Department of Pathology, University of Texas MD Anderson Cancer Center, Houston, TX 77030, USA; qgan@mdanderson.org (Q.G.); rwaters@mdanderson.org (R.E.W.); 4Department of Genomic Medicine, University of Texas MD Anderson Cancer Center, Houston, TX 77030, USA; lwang22@mdanderson.org

**Keywords:** SMARCA4, gastroesophageal adenocarcinoma, NGS panel

## Abstract

**Simple Summary:**

The clinical impact of SMARCA4 mutations (SMARCA4ms) in gastroesophageal adenocarcinoma (GEA) remains underexplored. We aimed to examine the association of SMARCA4ms with clinicopathological factors, patient survival, and co-occurrence with other gene mutations identified through an NGS panel in GEA patients. SMARCA4ms were identified in 19 out of 256 patients (7.4%). These SMARCA4ms were significantly associated with non-signet ring cell subtype (*p* = 0.044) and PD-L1 positive expression (*p* = 0.046), but not with other clinicopathological variables, including survival (*p* = 0.84). There were significant associations between SMARCA4ms and FANCA, IGF1R, KRAS, FANCL, and PTEN alterations. Notably, 15 of the 19 SMARCA4m cases involved SNV missense mutations, with frequent co-occurrences noted with TP53, KRAS, ARID1A, and ERBB2 mutations. These results serve as the first comprehensive examination of the relationship between SMARCA4ms and clinical outcomes in GEA.

**Abstract:**

Background: The clinical impact of SMARCA4 mutations (SMARCA4ms) in gastroesophageal adenocarcinoma (GEA) remains underexplored. This study aimed to examine the association of SMARCA4ms with clinical outcomes and co-occurrence with other gene mutations identified through a next-generation sequencing (NGS) panel in GEA patients. Methods: A total of 256 patients with metastatic or recurrent GEA who underwent NGS panel profiling at the MD Anderson Cancer Center between 2016 and 2022 were included. Comparative analyses were performed to assess clinical outcomes related to SMARCA4ms. The frequency and types of SMARCA4ms and their co-occurrence with other gene mutations were also examined. Results: SMARCA4ms were identified in 19 patients (7.4%). These SMARCA4ms were significantly associated with non-signet ring cell subtype (*p* = 0.044) and PD-L1 positive expression (*p* = 0.046). No difference in survival between the SMARCA4m and SMARCA4-normal group was observed (*p* = 0.84). There were significant associations between SMARCA4ms and FANCA, IGF1R, KRAS, FANCL, and PTEN alterations. Notably, 15 of the 19 SMARCA4m cases involved SNV missense mutations, with frequent co-occurrences noted with TP53, KRAS, ARID1A, and ERBB2 mutations. Conclusions: These results serve as the first comprehensive examination of the relationship between SMARCA4ms and clinical outcomes in GEA.

## 1. Introduction

Gastroesophageal adenocarcinoma (GEA) constitutes the main entity of upper gastrointestinal (GI) cancers which represent a significant global health burden. Esophageal cancer ranks seventh in incidence and sixth in mortality, and gastric cancer ranks fifth in incidence and fourth in mortality [1]. GEA is frequently diagnosed at an advanced stage, which often leads to poor prognosis. The cornerstone of treatment for advanced GEA is systemic therapy. Historically, cytotoxic agents, including fluoropyrimidines and platinum-based compounds, have been developed, demonstrating therapeutic efficacy. In recent years, the development of molecular-targeted drugs aimed at specific molecules such as human epidermal growth factor receptor-2 (HER2) and vascular endothelial growth factor receptor 2 (VEGFR2), along with immunotherapies targeting immune checkpoint molecules including programmed cell death 1 (PD-1) and programmed death-ligand 1 (PD-L1), has broadened the spectrum of treatment options [2,3,4,5,6,7]. The integration of conventional cytotoxic agents with these novel drugs has yielded promising therapeutic benefits. However, the issue of treatment resistance remains a critical consideration. For instance, the literature has documented that certain histological subtypes, including the diffuse type and signet ring cell carcinoma, exhibit substantial resistance to therapies [8,9]. Thus, characterizing individual tumor profiles is essential for identifying factors that contribute to treatment resistance and for exploring promising new therapeutic targets. This approach will ultimately enhance the optimization of treatment outcomes for patients with GEA. 

A deeper understanding of the genetic landscape of these cancers can shed light on the complex interplay between various genetic alterations and their roles in the development of cancer, potentially improving patient outcomes. Recent advances in genomic research have underscored the importance of genetic mutations in the pathogenesis and progression of these cancers [10,11]. A pivotal factor contributing to advancements in genomic research has been the introduction and widespread adoption of next-generation sequencing (NGS) technologies [12]. These innovative technologies have dramatically improved our ability to detect and analyze genetic mutations associated with various forms of cancer at a preliminary level. Furthermore, the expanding integration of NGS into clinical settings is anticipated to provide invaluable insights into the development of customized treatment strategies, thereby significantly enhancing the prospects of personalized medicine. For instance, assessing microsatellite instability (MSI) and tumor mutation burden (TMB) enables the utilization of tissue-agnostic therapies, such as immune checkpoint inhibitors [13]. The further clinical application of NGS holds the promise of transforming patient care by enabling more precise and personalized approaches to cancer treatment, tailored to individual genetic profiles [14,15].

The switch/sucrose non-fermentable (SWI/SNF) complex is an ATP-dependent chromatin remodeling subunit that plays a crucial role in regulating gene expression and maintaining genomic stability [16]. This complex comprises multiple genes, with mutations in these genes observed in approximately 20% of all human tumors, underscoring its significant role in cancer biology [17]. SMARCA4, also known as BRG1, encodes a core component of this complex and is recognized as a tumor suppressor gene. Studies have shown that mutations in SMARCA4 are observed in a wide variety of cancer types, highlighting its broad impact on oncogenesis [18,19,20]. In the context of upper gastrointestinal cancers, understanding the specific role and implication of this mutation remains a focal point of research. While a few studies have reported on undifferentiated carcinoma or rhabdoid tumors with SMARCA4 mutations in the gastrointestinal tract [21,22,23,24], the specific association between SMARCA4 mutations and clinicopathological factors or patient survival in GEA has not been fully elucidated. The scarcity of data on how SMARCA4 mutations interplay with other genetic aberrations in GEA further emphasizes the need for targeted research. Given the pivotal role of SMARCA4 mutations in tumorigenesis, investigating its mutations within the context of GEA has important clinical implications. Elucidating these mutations may identify new biomarkers for diagnosis, prognosis, or therapeutic targets, thereby transforming the management of these cancers. This highlights the urgent need for comprehensive research focused on the clinical significance of SMARCA4 mutations in GEA, with the aim of filling gaps in current understanding and optimizing patient care.

The primary objective of this study is to explore the association of SMARCA4 mutations with various clinicopathological factors and the patient survival rate in individuals diagnosed with GEA. In addition, we investigate the frequency and types of SMARCA4 mutations, as well as their relationships with other gene mutations detected in the NGS panel. By employing the results of the NGS panel for the precise determination of SMARCA4 mutations, this study aims to reveal the clinical characteristics of SMARCA4 mutations in patients with GEA and to gain deeper insights into the role of SMARCA4 within the broad genetic framework of GEA. This approach contributes to a more comprehensive understanding of its potential impact on disease progression and treatment outcomes.

## 2. Patients and Methods

### 2.1. Study Design and Patient Selections

This observational cohort study was conducted at the University of Texas, MD Anderson Cancer Center (MDACC), from September 2016 to April 2022. The primary objective was to examine the association between SMARCA4 mutations and clinicopathological factors, patient survival, and other gene mutations in GEA patients. In this study, we employed an NGS panel test as detailed below to identify mutations in the SMARCA4 gene. NGS panels are commonly utilized in clinical settings to detect genetic alterations in tumors, including point mutations and copy number variations. This approach is particularly relevant for cases of metastatic and recurrent disease where there is a need for systemic therapy, allowing for the assessment of additional systemic treatment options. Therefore, the eligibility criteria for study inclusion were a histologically confirmed diagnosis of metastatic or recurrent metastatic GEA patients who received genetic mutation testing using the institutional NGS panel. As for recurrent cases, all were subject to follow-up after radical surgery using imaging modalities, including computed tomography (CT) and esophagogastroduodenoscopy (EGD). When recurrence at a metastatic site was suspected with those image modalities, a biopsy was conducted, and recurrent metastasis was confirmed histologically. The physicians randomly selected patients for the NGS panel testing based on professional discretion, taking into account each patient’s background and disease course, when the NGS test was deemed likely to yield critical oncological insights. Patients were excluded if they had a cancer diagnosis other than adenocarcinomas, inadequate medical records for analyses, or if they did not have testing by an NGS panel. This study adhered to the ethical principles of the Declaration of Helsinki. The protocol was reviewed and approved by the MDACC Institutional Review Board (protocol number: PA12-1063), and all participants provided written informed consent for the use of their clinical data in the study.

### 2.2. Data Collection

The data extracted from medical records included comprehensive demographic details, such as age, sex, race, ethnicity, and performance status (PS), as assessed by the Eastern Cooperative Oncology Group (ECOG), in addition to the disease status, including initially diagnosed metastatic disease, developed metastases following progression after preoperative therapy, and recurrent metastatic disease after undergoing radical surgery. Furthermore, the data collection encompassed a wide range of tumor characteristics, including the exact location of the tumor within the GI tract, detailed histopathological classifications, and crucial biomarker information, including HER2, PD-L1, and MSI status, that could potentially inform therapeutic strategies. The statuses of HER2, PD-L1, and MSI were assessed using Immunohistochemistry (IHC). HER2 positivity was defined as either IHC 3+ or IHC 2+/FISH-positive. PD-L1 positivity was defined as a Combined Positive Score (CPS) greater than 1. MSI-High (MSI-H) was determined by the loss of expression of any of the DNA mismatch repair enzymes: MLH1, MSH2, MSH6, or PMS2. Survival information was meticulously gathered, covering the date of initial diagnosis, the date when metastasis was first identified, the date of the last follow-up, and if applicable, the date of death. These clinical data were systematically logged in our prospectively maintained clinical database. Moreover, outcomes from the NGS panel test described below were precisely documented, with special attention paid to the detection of the SMARCA4 mutation and the identification of additional mutations in other genes.

### 2.3. NGS Panel Testing

PCR-based sequencing was performed using an NGS platform on genomic DNA to screen for the somatic mutations of 134 genes and selected copy number variations (amplifications) in 47 genes (overlap: 146 genes total). The genomic reference sequence used was GRCh37/hg19. Mutations identified were described using an implementation of a standardized nomenclature developed by the Human Genome Variation Society (HGVS, https://hgvs-nomenclature.org/stable/ assessed on 15 March 2024). The gene list is provided in Appendix A. DNA extraction from the tissue samples was performed in the institutional diagnostics laboratory. The sensitivity of detection in this assay is partially determined by the depth of coverage, tumor percentage, and allelic frequency of the mutation. Although the NGS platform is capable of achieving significantly higher analytical sensitivity, for clinical purposes, the effective lower limit of detection (analytical sensitivity) for single nucleotide variants was determined to range from 5% (one mutant allele in the background of nineteen wild-type alleles) to 10% (one mutant allele in the background of nine wild-type alleles), taking into account the depth of coverage at specific bases and the ability to confirm low-level mutations through independent, conventional platforms. Since the sensitivity for amplifications depends on both input tumor percentage and the amplitude of gene amplification, a minimum of 20% tumor nuclei in the sample was required to minimize the risk of false-negative results. The analytical pipeline of this assay sought to normalize for differences in inter-amplicon performance and total sample loading, without attempting to adjust for tumor percentage. To ensure reporting was limited to high-confidence amplification calls, a nominal threshold of 7 for the estimated copy number was adopted. The sequencing coverage of the genes was determined by the adequacy of covered amplicons defined as those having a total coverage depth greater than or equal to 250 reads. The adequacy of coverage in this assay, across the full set of covered genes, exons, and codons for genes with exon-level sequencing, is described in Appendix A. A post-variant calling analysis and annotation tool, OncoSeek version 1.10.1.532, was used in the construction of the report. 

### 2.4. Statistical Analysis

Descriptive analyses were performed to outline the demographic and clinical characteristics of the study population. Categorical variables were analyzed using the chi-square test, and continuous variables were assessed using the *t*-test. Survival analysis was performed to evaluate the association between SMARCA4 mutations and patient survival. Kaplan–Meier curves for overall survival, defined as the time from the date of diagnosis to the event date (death from any cause) or the date of the last follow-up, were constructed. In addition, Kaplan–Meier curves for post-metastasis survival, defined as the period from the date of metastasis confirmation to the event date (death from any cause) or the date of the last follow-up, were also constructed. Difference in survival times was evaluated using the log-rank test. The association between SMARCA4 mutations and other gene mutations was assessed using Fisher’s exact test. A *p*-value of less than 0.05 was considered statistically significant in this study. All statistical analyses were conducted using R software (v 4.3.0).

## 3. Results

### 3.1. Patient Demographics and Disease Status

A total of 256 metastatic or recurrent GEA patients who underwent testing for the NGS panel were eligible in this study. Among these, 19 patients (7.4%) exhibited SMARCA4 mutations. Patient characteristics are outlined in Table 1. The mean ages in the SMARCA4-mutated and SMARCA4-normal groups were 60.2 and 56.1 years, respectively. The SMARCA4-mutated group comprised 14 men (73.7%) and 5 women (26.3%), while the SMARCA4-normal group included 171 men (72.2%) and 66 women (27.8%). There were no significant differences in age and sex distribution between the groups. Additionally, other demographic factors such as race, ethnicity, and ECOG-PS were comparable between the groups. In terms of disease status, among the SMARCA4-mutated group, 13 patients (68.4%) were initially diagnosed with metastatic disease, 2 patients (10.5%) developed metastases following progression after preoperative therapy, and 4 patients (21.1%) had recurrent metastatic disease after undergoing radical surgery. This pattern is comparable to the SMARCA4-normal group, where 146 patients (61.6%) were initially diagnosed with metastatic disease, 34 patients (14.3%) developed metastases due to progression after preoperative therapy, and 57 patients (24.1%) experienced recurrent disease at metastatic site following radical surgery. Overall, the demographic and disease status profiles were equivalent between the two groups.

### 3.2. Tumor Characteristics

Tumor characteristics for the SMARCA4-mutated and SMARCA4-normal groups are presented in Table 2. Regarding tumor location, in the SMARCA4-mutated group, eight (42.1%) tumors were located in the esophagus, five (26.3%) at the esophagogastric junction, and six (31.6%) in the stomach. In the SMARCA4-normal group, there were 64 (27.0%) tumors in the esophagus, 124 (52.3%) at the esophagogastric junction, and 49 (20.7%) in the stomach. Despite these distributions, the statistical analysis indicated no significant difference between the groups in terms of tumor location (*p* = 0.093). For histological type, the distribution in the SMARCA4-mutated group was as follows: well-differentiated 0 (0.0%), moderately differentiated 9 (47.4%), moderately to poorly differentiated 2 (10.5%), and poorly differentiated 8 (42.1%). On the other hand, in the SMARCA4-normal group, the counts were: well-differentiated 1 (0.4%), moderately differentiated 67 (28.3%), moderately to poorly differentiated 15 (6.3%), and poorly differentiated 154 (65.0%). Similar to the tumor location, no statistically significant difference was observed in histological type between the groups (*p* = 0.244). Notably, the frequency of the signet ring cell carcinoma, which is an unique subtype of adenocarcinoma, was significantly lower in the SMARCA4-mutated group than in the SMARCA4-normal group, with 3 cases (15.8%) versus 100 cases (42.2%); this difference was statistically significant (*p* = 0.044). Regarding biomarkers tested in routine clinical care, PD-L1 expression was significantly more prevalent in the SMARCA4-mutated group compared to the SMARCA4-normal group, observed in 16 cases (84.2%) versus 143 cases (60.3%), with this difference being statistically significant (*p* = 0.046). Conversely, the prevalence of HER2 was similar between the two groups. For MSI status, we observed that two cases (10.5%) in the SMARCA4-mutated group had MSI-H, a prevalence higher than seven cases (3.0%) in the SMARCA4-normal group. However, this difference did not reach statistical significance (*p* = 0.201).

### 3.3. Survival Analysis

We next performed survival analysis to examine the impact of SMARCA4 mutation on patient survival. The median follow-up duration was 1.37 years. During this period, survival events were recorded in 180 out of 256 cases. The Kaplan–Meier curve illustrating the overall survival for both the SMARCA4-mutated and SMARCA4-normal groups is depicted in Figure 1a. The median overall survival was 2.12 years for the SMARCA4-mutated group and 1.69 years for the SMARCA4-normal group. No significant difference in survival probability was observed between the two groups (*p* = 0.84). Furthermore, the Kaplan–Meier curve for post-metastasis survival for both SMARCA4-mutated and SMARCA4-normal groups is presented in Figure 1b. The median post-metastasis survival was 1.71 years for the SMARCA4-mutated group and 1.25 years for the SMARCA4-normal group. Similar to the overall survival findings, there was no significant difference in survival probability between the two groups (*p* = 0.31). Taken together, these findings suggest that SMARCA4 mutations exert minimal impact on patient survival.

### 3.4. Genotypic Landscape of SMARCA4 Mutations Based on the NGS Panel

Finally, we investigated the frequency and types of SMARCA4 mutations, along with their co-occurrence with other gene mutations detected by the NGS panel. Within our study cohort, the frequency of SMARCA4 mutations ranked seventh in frequency, following mutations in TP53, KRAS, ARID1A, ERBB2, PIK3CA, and FGFR2, which are detailed in Figure 2a. Notably, significant associations were observed between SMARCA4 mutations and mutations in FANCA, IGF1R, KRAS, FANCL, and PTEN, which are presented in Figure 2b. Further detailed in Figure 2c is a depiction of the various types of genetic alterations alongside the genes that were found to be co-mutated within the SMARCA4-mutated cases in our study. The total number of genetic mutations in these cases ranged from 2 to 15, interestingly with no instances where SMARCA4 mutations were found to occur in isolation. Among the 19 cases analyzed, 15 exhibited single nucleotide variant (SNV) missense mutations. Mutations in TP53, KRAS, ARID1A, and ERBB2 were identified as those most frequently co-occurring with SMARCA4 mutations, underscoring potential patterns of genetic interplay significant to the understanding of the disease.

## 4. Discussion

This observational cohort study via an NGS panel revealed that 7.4% of GEA patients harbored SMARCA4 mutations. This prevalence is consistent with findings reported in the previous literature, underscoring the reliability of NGS panels in clinical diagnostics [16]. Notably, we demonstrated that SMARCA4 mutations were significantly associated with non-signet ring cell subtype, as well as PD-L1 positive expression, which has not been extensively documented in GEA. Contrary to expectations, SMARCA4 mutations did not significantly affect survival. Furthermore, employing a comprehensive NGS panel enabled us to clarify the connections between SMARCA4 gene mutations and various other genetic mutations. The significant strength of this study lies in its investigation of the association between SMARCA4 mutations and the clinical information of GEA within a real-world context, examining the largest number of GEA cases reported to date. Utilizing a comprehensive NGS panel, which is employed in clinical practice, has not only allowed us to determine the frequency and types of SMARCA4 mutations but also to uncover information on covariations with other genetic mutations. This approach has notably enhanced our understanding of the molecular interactions and clinical relevance of SMARCA4 mutations in GEA, contributing to a deeper insight into the molecular characterization of this cancer type.

The exploration of SMARCA4 mutations in GEA is relatively uncharted in the literature, with few studies addressing their clinicopathological implications [25,26]. Our study contributes to this emerging field by demonstrating a significant association between SMARCA4 mutations and the non-signet ring cell subtype. However, these findings remain inconclusive, partly due to the rarity of SMARCA4-altered gastric cancer and its inherent intratumoral heterogeneity and histomorphological diversity [27]. Regarding survival, the literature on various cancer types suggests a link between SMARCA4 mutations or loss of function and poor prognosis [28]. However, the evidence specific to GEA is sparse and inconsistent. Schallenberg et al. reported that SMARCA4 loss of function was not correlated with shortened overall survival in esophageal adenocarcinoma [29], while Zhang et al. observed that the SMARCA4-lost group correlates with worse prognosis compared to the corresponding SMARCA4-present groups [30], implying a divergence in research regarding survival implications of the mutations in GEA. In addition, these studies primarily assessed SMARCA4 through immunohistochemistry (IHC), differing from our NGS panel approach. Our finding of no significant correlation between SMARCA4 mutations and survival in GEA could be attributed to the absence of comprehensive data on other SWI/SNF-related genes. This highlights the necessity for further research involving these additional genes in a larger cohort to fully understand the role of SMARCA4 mutations in the clinical trajectory of GEA.

In our NGS panel analysis, we found that SMARCA4 mutations, occurring in 7.4% of cases, were the seventh most frequent type of genetic alteration in our cohort. The predominant mutation type was SNV missense mutations, a pattern consistent with observations in other cancer types [20]. Our study also demonstrated significant associations between SMARCA4 mutations and alterations in several genes, including FANCA, IGF1R, KRAS, FANCL, and PTEN. These findings suggest the presence of a complex network of genetic interactions that could potentially impact the tumorigenic process or progression of GEA. Furthermore, SMARCA4 mutations were typically found to co-occur with mutations in well-known oncogenes and tumor suppressor genes such as TP53, KRAS, ARID1A, and ERBB2, rather than appearing in isolation. Pan et al. conducted an analysis of SMARCA4 mutations across various cancers using an NGS platform, similar to our approach [31]. They identified that the most frequent co-mutated genes with SMARCA4 were TP53, KRAS, CDKN2A, STK11, and Keap1. Unlike their study, our NGS panel did not include Keap1, and we did not identify any cases with STK11 mutations. Nevertheless, similar to their findings, we observed frequent co-mutations of TP53, KRAS, and CDKN2A with SMARCA4 mutations. Given the relatively low frequency of SMARCA4 mutations, there is a compelling need for further exploration of these co-mutations within a larger cohort of GEA patients. Investigating SMARCA4 mutations in more extensive cohorts of GEA could provide a more thorough understanding of the interplay between SMARCA4 mutations and other genetic alterations. Additionally, considering the functional roles of these co-occurred genes alongside SMARCA4’s function, further research is desired to unravel the comprehensive impact of these genetic interactions on cancer pathogenesis within the context of GEA. This approach will not only enrich our comprehension of their collective impact but also potentially identify novel therapeutic targets.

Attempts to target SMARCA4 in various types of tumors, including non-small-cell lung cancer and brain tumors, have been reported in the preclinical phase [32,33]. SMARCA4 and SMARCA2 function as mutually exclusive catalytic subunits in the SWI/SNF complex. Since SMARCA4 is generally regarded as a tumor suppressor gene and SMARCA4 mutations are thought to be involved in cancer progression, SMARCA2 targeting and the induction of cell death using the concept of synthetic lethality may be considered in the treatment of SMARCA4-mutated cancers. This effectively arrests the activity of the SWI/SNF chromatin remodeling complex and inhibits cancer cell growth. Although reports on gastroesophageal adenocarcinoma (GEA) in particular are limited, the mechanism of action is promising and may be applied to GEA in the future. Additionally, there are intriguing reports concerning the relationship between SMARCA4 mutations and the immune microenvironment. Various studies have noted an association between SMARCA4 mutations and an increase in tumor-infiltrating immune cells, as well as T cell function [34,35,36,37]. Notably, previous research found that SMARCA4 mutations were linked to TMB, MSI, and dMMR [27,38], all of which have demonstrated efficacy in response to immunotherapy [39,40,41]. Our research also implied associations between SMARCA4 mutations and MSI-H, though this difference did not reach statistical significance. Moreover, we also observed a significant association between SMARCA4 mutations and PD-L1 expression. Given these insights, not only treatments targeting SMARCA4/SMARCA2, but also immunotherapy may be considered as a potential therapeutic strategy for cases with SMARCA4 mutations. In fact, there have been reports suggesting the potential of immunotherapy for ARID1A loss, which encodes the same SWI/SNF unit [42,43]. Further comprehensive research, focusing on GEA cases, is warranted to explore and validate the role of immunotherapy in this context [44].

The limitations of this study primarily arise from its observational nature, which inherently carries the risk of selection bias, especially considering that the patient cohort was exclusively sourced from a single institution. This limitation potentially restricts the generalizability of our findings to broader populations. Additionally, while our analysis revealed significant associations between SMARCA4 mutations and various clinical and molecular parameters, we did not delve into the intricate molecular mechanisms driving these relationships. This gap underscores the imperative need for subsequent functional studies aimed at unraveling the biological underpinnings of the observed correlations. Such investigations are crucial for confirming the causal relationships and understanding the role of SMARCA4 mutations in the disease process. Furthermore, expanding the scope of research to include other gene clusters responsible for encoding different subunits of the SWI/SNF complex would provide a more comprehensive understanding of the complex’s overall role in cancer biology. Exploring these additional gene clusters could unveil new dimensions of how alterations in the SWI/SNF complex contribute to tumorigenesis and progression, offering valuable insights into potential therapeutic targets within this crucial area of cancer research.

## 5. Conclusions

In conclusion, this study represents a pioneering effort in elucidating both the clinical and genotypic landscape of SMARCA4 mutations in GEA using an NGS panel. Our findings offered the first comprehensive insight into how SMARCA4 mutations correlate with clinicopathological factors, their co-occurrence with other gene mutations, and their impact on patient survival in GEA. By leveraging the advanced capabilities of NGS, we have provided a more nuanced understanding of the role of SMARCA4 in GEA, setting a foundation for future research and potential therapeutic strategies. This study marks a significant step forward in the field of GEA research, paving the way for more targeted and effective treatment approaches based on genetic profiling.

## Figures and Tables

**Figure 1 cancers-16-01300-f001:**
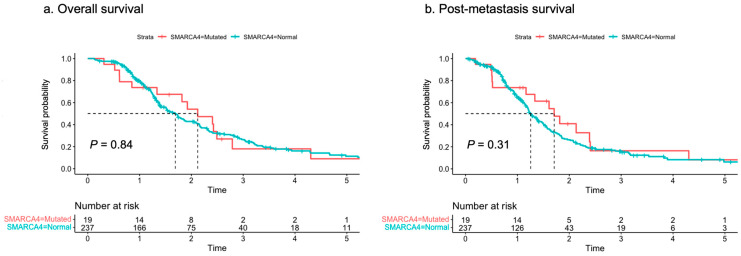
(**a**) Kaplan–Meier curves for overall survival in SMARCA4-mutated and SMARCA4-normal groups. (**b**) The Kaplan–Meier curves for post-metastatic survival in SMARCA4-mutated and SMARCA4-normal groups. Difference in survival times was evaluated using the log-rank test.

**Figure 2 cancers-16-01300-f002:**
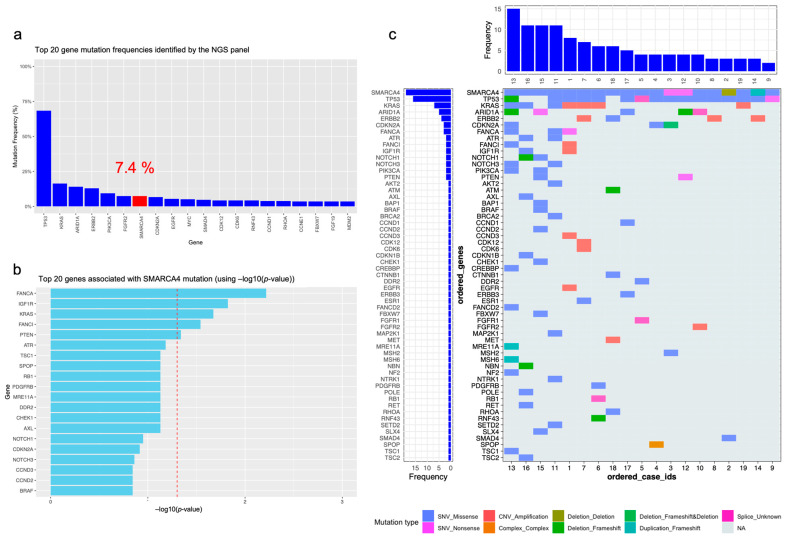
(**a**) Top 20 gene mutation frequencies identified through the NGS panel. (**b**) Top 20 genes associated with SMARCA4 mutations. (**c**) Color map of mutation types and frequencies for SMARCA4 and co-occurring gene mutations in patients with SMARCA4 mutations.

**Table 1 cancers-16-01300-t001:** Patient demographics in SMARCA4-mutated vs. SMARCA4-normal cases.

Variables	Overall (*n* = 256)	SMARCA4-Mutated (*n* = 19)	SMARCA4-Normal (*n* = 237)	*p*-Value
Age				0.184
Mean (SD)	56.4 (13.0)	60.2 (14.2)	56.1 (12.8)	
Sex, *n* (%)				1
Male	185 (72.3)	14 (73.7)	171 (72.2)	
Female	71 (27.7)	5 (26.3)	66 (27.8)	
Race, *n* (%)				0.597
Asian	20 (7.8)	1 (5.3)	19 (8.0)	
Black or African American	17 (6.6)	1 (5.3)	16 (6.8)	
White	202 (78.9)	17 (89.5)	185 (78.1)	
Other Race	17 (6.6)	0 (0.0)	17 (7.2)	
Ethnicity *n* (%)				0.907
Hispanic or Latino	44 (17.2)	3 (15.8)	41 (17.3)	
Not Hispanic or Latino	210 (82.0)	16 (84.2)	194 (81.9)	
N/A	2 (0.8)	0 (0.0)	2 (0.8)	
ECOG-PS, *n* (%)				0.607
0	95 (37.1)	6 (31.6)	89 (37.6)	
1	146 (57.0)	13 (68.4)	133 (56.1)	
2	13 (5.1)	0 (0.0)	13 (5.5)	
3	2 (0.8)	0 (0.0)	2 (0.8)	
Disease status, *n* (%)				0.827
Initially metastatic disease	159 (62.1)	13 (68.4)	146 (61.6)	
Metastasis after preoperative therapy	36 (14.1)	2 (10.5)	34 (14.3)	
Recurrent metastatic disease	61 (23.8)	4 (21.1)	57 (24.1)	

ECOG-PS = Eastern Cooperative Oncology Group Performance Status; N/A = Not available.

**Table 2 cancers-16-01300-t002:** Tumor characteristics in SMARCA4-mutated vs. SMARCA4-normal cases.

Variables	Overall (*n* = 256)	SMARCA4-Mutated (*n* = 19)	SMARCA4-Normal (*n* = 237)	*p*-Value
Tumor location, *n* (%)				0.093
Esophagus	72 (28.1)	8 (42.1)	64 (27.0)	
GEJ	129 (50.4)	5 (26.3)	124 (52.3)	
Stomach	55 (21.5)	6 (31.6)	49 (20.7)	
Histological type, *n* (%)				0.244
Well-differentiated	1 (0.4)	0 (0.0)	1 (0.4)	
Moderately differentiated	76 (29.7)	9 (47.4)	67 (28.3)	
Moderately to poorly differentiated	17 (6.6)	2 (10.5)	15 (6.3)	
Poorly differentiated	162 (63.3)	8 (42.1)	154 (65.0)	
Signet ring cell component, *n* (%)				0.044
Yes	103 (40.2)	3 (15.8)	100 (42.2)	
No	153 (59.8)	16 (84.2)	137 (57.8)	
HER2, *n* (%)				0.744
Positive	46 (18.0)	4 (21.1)	42 (17.7)	
Negative	204 (79.7)	15 (78.9)	189 (79.7)	
N/A	6 (2.3)	0 (0.0)	6 (2.5)	
PD-L1, *n* (%)				0.046
Positive	159 (62.1)	16 (84.2)	143 (60.3)	
Negative	78 (30.5)	1 (5.3)	77 (32.5)	
N/A	19 (7.4)	2 (10.5)	17 (7.2)	
MSI status, *n* (%)				0.201
MSI-H	9 (3.5)	2 (10.5)	7 (3.0)	
MSS	226 (88.3)	15 (78.9)	211 (89.0)	
N/A	21 (8.2)	2 (10.5)	19 (8.0)	

GEJ = Gastroesophageal junction; HER2 = Human epidermal growth factor receptor 2; PD-L1 = Programmed Cell Death Ligand 1; MSI = Microsatellite instability; MSI-H = Microsatellite instability-high; MSS = Microsatellite stable; N/A = Not available.

## Data Availability

The data presented in this study are available upon request from the corresponding author, subject to ethical restrictions related to patient privacy concerns.

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
