# Peer review of "SMARCA4 Mutations in Gastroesophageal Adenocarcinoma: An Observational Study via a Next-Generation Sequencing Panel"

_cancers, 2024, doi:10.3390/cancers16071300_

Round 1

Reviewer 1 Report

Comments and Suggestions for Authors

In this study, the authors evaluated by using Next-Generation Sequencing (NGS), the association of SMARCA4 mutations (SMARCA4m) with clinicopathological factors, their relationship with other gene mutations, and survival of patient with metastatic or recurrent gastroesophageal adenocarcinoma (GEA).

They showed that SMARCA4m associate with non-signet ring cell subtype and PD-L1 positive expression but not with patient survival. A significant association between SMARCA4m and FANCA, IGF1R, KRAS, FANCL, and PTEN alterations was also found.

Overall, this is an interesting and well written study. However, there are some points that need attention.

1) In the Patients and Methods section, the authors state that the study population was patients with “confirmed diagnosis of metastatic or recurrent GEA” (lines 80-81) and that “The physician randomly selected patients for the NGS testing, according to their professional discretion.” (lines 82-83).

In my opinion the selection criteria need to be clear, well defined and explained.

How did they confirm the diagnosis, histologically? By imaging?

Why they included only cases of metastatic or recurrent disease?

How did they define recurrent disease?

What do they mean by “professional discretion”?

2) In the results section, there are data on patients with an initially unresectable disease, patients who became unresectable after induction therapy, and patients with recurrent disease (lines 129-135).

Unresectable might be tumors due to local invasion and do not necessarily imply metastatic disease.

“Recurrent disease”. What do they mean? Recurrence after surgery? Disease progression after neoadjuvant chemotherapy?

3) A minor point. Figure 2 instead of Figure 1 (line 165).

Reviewer 2 Report

Comments and Suggestions for Authors

This is a well-written and easy to read manuscript regarding the detection of SMARCA4 mutations in GEA.

The only part of the manuscript that needs to be improved is the description of the NGS assay used. There is not enough information provided neither on the assay's characteristics nor on the analysis performed. Details on the NGS runs would also be useful to have. An orthogonal assay is mentioned without any further information (type of the assay, number of samples assayed etc).

Overall, however, I think that the findings of this research project are not significant enough to be published in Cancers. After revision of the Materials and Methods Section, I would recommend submission to a journal like Diagnostics.

Reviewer 3 Report

Comments and Suggestions for Authors

It is not clear why is worthy and important to test the  SMARCA4 mutation status in astroesophageal Adenocarcinoma. The clinical significance of the study should be highlighted.

Control group is not defined

Which is the innovation of the findings apart of the use of NGS?

Round 2

Reviewer 3 Report

Comments and Suggestions for Authors

I do not find the manuscript suitable for high impact journal. It is a descriptive manuscript based on one technique NGS